# Human Serum Betaine and Associated Biomarker Concentrations Following a 14 Day Supplemental Betaine Loading Protocol and during a 28 Day Washout Period: A Pilot Investigation

**DOI:** 10.3390/nu14030498

**Published:** 2022-01-24

**Authors:** Steven B. Machek, Emilia E. Zawieja, Jeffery L. Heileson, Dillon R. Harris, Dylan T. Wilburn, Emma A. Fletcher, Jason M. Cholewa, Artur Szwengiel, Agata Chmurzynska, Darryn S. Willoughby

**Affiliations:** 1Department of Health, Human Performance, and Recreation, Robbins College of Health and Human Sciences, Baylor University, Waco, TX 76706, USA; smachek@ozarks.edu (S.B.M.); Jeffery_heileson@baylor.edu (J.L.H.); dillon_harris1@baylor.edu (D.R.H.); Dylan_wilburn1@baylor.edu (D.T.W.); emma_fletcher1@baylor.edu (E.A.F.); 2Division of Natural Sciences and Mathematics, University of the Ozarks, Clarksville, AR 72830, USA; 3Department of Human Nutrition and Dietetics, The Poznań University of Life Sciences, 60-637 Poznań, Poland; emilia.zawieja@up.poznan.pl (E.E.Z.); agata.chmurzynska@up.poznan.pl (A.C.); 4Department of Biology, College of Arts and Sciences, Baylor University, Waco, TX 76706, USA; 5Exercise Physiology Department, University of Lynchburg, Lynchburg, VA 24501, USA; Cholewa_jm@lynchburg.edu; 6Institute of Food Technology of Plant Origin, The Poznań University of Life Sciences, 60-637 Poznań, Poland; Artur.szwengiel@up.poznan.pl; 7School of Exercise and Sport Science, Mayborn College of Health Sciences, University of Mary Hardin-Baylor, Belton, TX 76513, USA

**Keywords:** betaine, trimethylglycine, washout, serum, homocysteine, insulin-like growth factor-1 (IGF-1), growth hormone (GH), hematocrit, intracellular water (ICW), extracellular water (ECW)

## Abstract

Several previous investigations have employed betaine supplementation in randomized controlled crossover designs to assess its ostensible ergogenic potential. Nevertheless, prior methodology is predicated on limited pharmacokinetic data and an appropriate betaine-specific washout period is hitherto undescribed. The purpose of the present pilot investigation was therein to determine whether a 28 day washout period was sufficient to return serum betaine concentrations to baseline following a supplementation protocol. Five resistance-trained men (26 ± 6 y) supplemented with 6 g/day betaine anhydrous for 14 days and subsequently visited the lab 10 additional times during a 28 day washout period. Participants underwent venipuncture to assess serum betaine and several other parameters before (PRE) and periodically throughout the washout timeframe (POST0, -4, -7, -10, -13, -16, -19, -22, -25 and -28). All analyses were performed at a significance level of *p* < 0.05. While analyses failed to detect any differences in any other serum biomarker (*p* > 0.05), serum betaine was significantly elevated from PRE-to-POST0 (*p* = 0.047; 2.31 ± 1.05 to 11.1 ± 4.91 µg·mL^−1^) and was statistically indistinguishable from baseline at POST4 (*p* = 1.00). Nevertheless, visual data assessment and an inability to assess skeletal muscle concentrations would otherwise suggest that a more conservative 7 day washout period is sufficient to truly return both serum-and-skeletal muscle betaine content to pre-supplementation levels.

## 1. Introduction

Betaine, or *N*-*N*-*N*-trimethylglycine, is a product of choline metabolism and can be found in foods including sugar beets, wheat bran, spinach, and shrimp [1]. It primarily functions as a methyl donor via the enzymatic action of betaine-homocysteine methyltransferase (BHMT), complementing the roles of folate-and-cyanocobalamin (vitamin B12) metabolism to transmethylate homocysteine (HCY) to methionine [2,3]. Elevated plasma HCY levels (hyperhomocysteinemia) are highly correlated to atherosclerosis, potentially contributing to oxidative stress and downstream vascular function impairments [4]. Untreated hyperhomocysteinemia via insufficient methyl donors may result in the conversion of homocysteine to its thiolactone (HCTL) analogue, which contains a high-energy thioester bond that can inhibit proper insulin and growth factor signaling [5]. Betaine supplementation also facilitates synthesis of the universal methyl donor, s-adenosyl methionine (SAM), via potentiating methionine metabolism, the former of which is involved in DNA methylation and thus modulating gene expression, as well as neurotransmitter, creatine, and catecholamine synthesis [3]. Lastly, this compound is further reported to increase red blood cell and hemoglobin counts commensurate with a subsequently augmented hematocrit percentage, potentially due to its role in facilitating the folate-dependent synthesis of purines and pyrimidines [6,7]. Betaine therein clearly demonstrates versatility in fostering various physiological functions.

Although relatively little data exist within an ergogenic context, betaine supplementation has a potentially understated role in augmenting exercise performance [8]. This compound not only facilitates normal methylation and the prevention of HCTL accumulation, but can augment anabolic signaling through mechanisms including increased growth hormone (GH) secretion and subsequently elevated insulin-like growth factor-1 (IGF-1) concentrations, alongside potentiating both upstream and downstream mammalian target of rapamycin complex 1 (mTORC1) signaling [1,5]. Betaine is typically supplemented in doses between 2.5 and 6.0 g per day, usually split between two equivalent doses [9]. Very little research has detailed the effects of betaine supplementation on performance and body composition; however, it appears that it primarily augments muscular endurance via increasing overall volume in higher (8–15) repetition ranges [8]. Other positive data needing further support include increased skeletal muscle cross-sectional area commensurate with increased training capacity and reductions in fat mass, as well as enhanced peak power and isometric strength indices [10,11,12,13,14]. Notwithstanding the aforementioned methyl metabolism enhancements, several betaine-associated performance benefits may be mediated through its role as an organic osmolyte, stabilizing intracellular protein structures (similar to molecular chaperones) and attenuating osmotic stress after being taken up by the osmoregulated betaine/γ-aminobutyric acid transporter (BGT-1; the predominant betaine channel in humans) [5,15,16]. Although this compound classically protects cells from cell shrinkage-mediated apoptosis in hypertonic environments such as the renal medulla, it has been postulated that it may also relieve glycolytic exercise-mediated hypotonic stress, which is characterized by short rest periods and concomitant localized acidosis [9,15,17,18].

Despite the increasing use of betaine as an ergogenic supplement, relatively little research has been performed determining its pharmacokinetics in human subjects [8]. The elimination half-life of betaine is highly variable with acute doses averaging approximately 14 ± 7 h [19,20]. This timeframe is dramatically augmented with consistent dosing, showing that a five-day supplementation protocol increased the plasma elimination half-life to an average of approximately 41 h [20]. Multiple doses did not significantly affect time to maximum plasma concentrations (i.e., T_max_; average ~1 h), but did increase peak serum concentrations (i.e., C_max_) from an average of 0.939 to 1.456 mmol/L. Interestingly, the evidence regarding tissue-specific saturation is sparse; however, a past rodent model demonstrates peak skeletal muscle concentrations occur at approximately 10 days with levels slowly increasing up to 56 days following betaine administration [21]. While serum concentrations do not necessarily reflect tissue concentrations, there appears to be a strong correlation between serum and skeletal muscle concentrations that are not mirrored in many other tissue types [22]. A further human investigation saw increased serum betaine following 8 weeks of supplementation, which continued to slowly increase up to the last sampling point at 24 weeks [23]. While it can be noted that peak skeletal muscle-specific betaine concentrations can be reached within 2 weeks—notwithstanding the potential for further slow, progressive increases—performance-oriented washout times in the current literature are inconsistent. Washout periods are commonly employed in randomized crossover designs, serving to reverse treatment impacts and ensure baseline comparability before receiving the alternative intervention [24]. This approach is especially valuable in exercise science research, producing sufficient statistical power with fewer participants in study designs that are often laborious and limited by resources [24,25]. Nevertheless, the available betaine supplementation literature commonly uses a wash-out period anywhere between one to three weeks, but no previous investigations have determined an appropriate post-supplementation washout timeframe and thus warrants investigation [9]. The current study therefore aimed to test the hypothesis that a 28 day washout period was sufficient to discernably return serum betaine concentrations and several associated parameters (total body water indices, hematocrit, as well as serum GH, IGF-1, and HCY) to baseline levels following a 14 day supplementation period. Notably, our group recruited a small participant pool designed as a pilot investigation, seeking to elucidate an appropriate timeline for a future crossover investigation employing betaine supplementation.

## 2. Materials and Methods

### 2.1. Participants

Five apparently healthy, recreationally resistance-trained (modified from the American College of Sports Medicine recommendations; ≥30 min exercise, ≥3 days per week, over the last 3 months) men between the ages of 18 and 35 years volunteered to serve as subjects in this study as a convenience sample [26]. Enrollment was open to men of all ethnicities. Women were not recruited due to sex-specific variations in methyl metabolism and tissue betaine concentrations [13,22]. The use of blood thinning (e.g., Warfarin and Jantoven), heart, pulmonary, thyroid, antihypertensive, anti-hyperlipidemic, hypoglycemic, endocrinologic (e.g., prednisone, Ritalin, and Adderall), or neuromuscular/neurological medications was further prohibited for eligibility. Only participants considered low risk for cardiovascular disease with no contraindications to exercise as outlined by the ACSM and have not consumed any nutritional supplements (excluding multivitamins) one month prior to this study were allowed to participate. All eligible subjects signed university-approved informed consent documents and approval was granted by the Institutional Review Board for Human Subjects at Baylor University (reference #1618928, approval date: 9 August 2020). In addition, all experimental procedures involved in this study conformed to ethical considerations of the Helsinki Code.

### 2.2. Experimental Approach and Betaine Supplementation

Participants visited the laboratory on 12 separate occasions following an overnight fast, including an initial screening/familiarization visit, a second visit to assess baseline parameters (PRE) and provide each participant their supplement, and 10 post-supplementation washout assessment visits (see Figure 1). Specifically, participants consumed 3 g twice daily (separated by ~12 h) betaine anhydrous (Vital Pharmaceuticals [VPX] Inc., Weston, FL, USA) for 14 days. This loading regimen was chosen to represent the highest betaine dose administered in the performance literature [9]. Furthermore, the present supplementation timeframe was selected to ostensibly ensure peak skeletal muscle betaine concentrations, predicated on prior rodent pharmacokinetic data [21]. To assess the lasting impact of betaine loading on various serum and hydration-associated parameters, participants additionally visited the laboratory the day following their supplementation period (POST0), as well as 4, 7, 10, 13, 16, 19, 22, 25, and 28 days post-supplementation (POST-4,-7,-10,-13,-16,-19,-22,-25 and -28, respectively). A 28 day washout period was specifically chosen to fully encompass the existing range of washout periods previously reported in the betaine-specific performance literature (1–3 weeks), whilst also anticipating that the relatively larger dose within our loading regimen may prolong elevated serum betaine concentrations after ceasing supplementation [9,10,11,27]. All participants were asked to consume their last supplement dose 12 h from the POST0 visit to ostensibly better standardize serum parameters. Supplement compliance was confirmed by the lead research technician and demonstrated by the return of an empty supplement container on the POST0 visit, alongside twice weekly verbal confirmation of protocol adherence via telephone communication.

### 2.3. Anthropometrics, Whole-Body Hydration, and Body Composition Analysis

Height (cm) was determined on a standard dual beam balance scale (Detecto, Bridgeview, IL, USA) during the PRE visit. Furthermore, participant weight was similarly assessed, alongside intracellular and extracellular water (ICW and ECW, respectively) on the PRE and every post-supplementation visit. Cellular water compartments were determined with a multifrequency bioelectrical impedance analysis (BIA; InBody 570, Cerritos, CA, USA) device using a low energy, high frequency current [28]. The InBody analyzer utilizes eight tactile electrodes (two in contact with the palm and thumb of each hand and two with the anterior and posterior aspects of the sole of the foot) and has been validated for accurate determination of extra (ECW) and intracellular water (ICW) content in multiple demographics [29,30]. Lastly, percent body fat (BF%), fat mass, and fat free mass were determined on the PRE, POST0, and POST28 visits, using dual-energy-X-ray-absorptiometry (DEXA) (Hologic Discovery Series W, Waltham, MA, USA). Quality control calibration procedures were performed on a spine phantom (Hologic X-CAIBER Model DPA/QDR-1 anthropometric spine phantom) and a density step calibration phantom prior to each testing session. All participants were analyzed via DEXA wearing minimal clothing, in an overnight fasted state, and after avoiding strenuous exercise for at least 48 h prior to assessment.

### 2.4. Dietary Tracking and Records

Prior to each participant’s PRE, POST0, POST7, POST13, POST22, and POST28 visits, they were required to record 24 h food recalls via the MyFitnessPal (San Francisco, CA, USA) application, which were subsequently transferred and analyzed for micronutrient intake via Food Processor dietary assessment program (ESHA Research, Salem, OR, USA). Participants were instructed on how to use relevant features if unfamiliar with the modality. Moreover, our board-certified laboratory Registered Dietitian (RD) assessed the logs for any apparent nutrient inadequacies that may impede HCY metabolism (folate, as well as vitamins B2, B6, B12, and choline) [7]. If participants demonstrated an insufficient intake of the aforementioned nutrients based on current American dietary standards, they were provided a B-complex multivitamin (Nature Made Super B Energy Complex Softgels, West Hills, CA, USA; thiamin 1.5 mg, riboflavin 1.7 mg, niacin 20 mg, pyridoxine hydrochloride 2 mg, folic acid 400 mcg, cyanocobalamin 6 mg, biotin 300 mcg, and pantothenic acid 10 mg) in a quantity commensurate to the number of days leading up until their next dietary assessment visit and instructed to take one softgel per day alongside their first daily betaine dose [31]. Diets were reassessed at each 24 h recall visit to determine whether the B-complex would remain necessary until the following assessment. Lastly, it is important to note that participants’ diets were not standardized, but they were asked not to change their dietary habits during the study duration. Dietary macronutrient (protein, carbohydrate, and fat) as well as fiber intake were calculated on all 24 h dietary records. All macronutrient and fiber data were subsequently normalized to weight (kg) of that respective visit for further statistical analyses.

### 2.5. Venipuncture

Venous blood samples were obtained in 10 mL vacutainer tubes using a 21-gauge phlebotomy needle inserted into the antecubital vein. Blood samples were allowed to stand at room temperature for 10 min and then centrifuged at 2500 rpm for 15 min. The serum was then removed and immediately frozen at −80 °C for later analysis. Eleven blood samples were obtained during the course of this study. The blood samples were collected at the PRE and every post-supplementation visit. Moreover, all blood samples were subsequently and immediately assessed for hematocrit, whereby whole blood was drawn into micro-hematocrit tubes by capillary action and sealed with clay material [32]. These tubes were then spun for 2 min before removing and subsequently analyzed on a hematocrit reader card. Normal ranges for adult males were considered between 42 and 52% packed cell volume (PCV%) [32].

### 2.6. Serum Betaine Analysis

Serum samples were further processed in their respective 1.5 mL Eppendorf tubes, adding plasma (25 µL), internal standards (12.5 ng in 25 µL of water), as well as 300 µL of cold acetonitrile, and were vortexed for 15 s. Sample protein precipitates were stored for 0.5 h at 4 °C and then the mixtures were centrifuged (15 min, 20,000× *g*, 4 °C). The supernatant was transferred to the vial and analyzed by liquid chromatography-mass spectrometry (LC-MS).

Serum betaine was assessed via LC-MS as described previously [33]. The isotope dilution analysis was performed and this approach was detailed and formerly evaluated by Koc et al. [34]. Furthermore, betaine was employed as the internal standard. The chromatographic separation of analyses was achieved using Cogent 4 µm Diamond Hybride column 150 × 2.1 mm (Microsolv Technology Corporation, Leland, NC, USA) and ultra high-performance liquid chromatography (Dionex UltiMate 3000 UHPLC, Thermo Fisher Scientific, Sunnyvale, CA, USA) coupled to a Bruker maXis impact ultrahigh resolution orthogonal quadrupole-time-of-flight accelerator (qTOF) equipped with an ESI source and operated in the positive-ion mode (Bruker Daltonik, Bremen, Germany).

The mobile phase was comprised of water containing 5 mM ammonium formate in acetonitrile water (95:5 *v*/*v*) (A) and 5 mM ammonium formate in water (B). The flow rate was 0.4 mL/min, the gradient changed 0–7 min, 17–25% B, 7–12 min, 25–65% B; 12–16 min, 65% B; the column was re-equilibrated with 17% solvent B for 6 min. The cleaning of column with methanol:water (1:1, *v*/*v*) was performed every 10 injections for 5 min. The syringe and needle were washed with methanol:water (1:1, *v*/*v*) before and after injection (10 µL) of sample. The carryover between samples was not observed. The column temperate was set to 40 °C. The ESI-MS settings were as follows: capillary voltage 4500 V, nebulizing gas 2.0 bar, and dry gas 11 L/min at 220 °C. The scan range was from mass-to-charge ratio (*m*/*z*) 50–800. The ESI system was calibrated using sodium lithium formate ion clusters. Molecular ions were extracted from full scan chromatograms (±0.005 *m*/*z*) and peak areas were integrated with TASQ 2.1 (Bruker Daltonik, Bremen, Germany). The compounds present in each sample were identified based on retention time of standard and isotope information from the MS detector. Analyte free matrix was not available. Calibration and quality control (QC) samples were prepared in water as surrogate matrix. It was reported that calibration standards and QC prepared in water can be used instead of matrix-matched calibration and controls [35]. Recovery of QC was higher than 95% and recovery of standards spiked to serum samples was above 92%. The standards were used for calibration by plotting the ratio of the analyte signal to the adequate ISTD signal (relative response –y) as a function of relative concentration –x (ratio of analyte concentration to ISTD concentration) Coefficient of determination (R^2^) for all calibration curves was higher than 0.99.

### 2.7. Serum Growth Hormone (GH), Insulin-like Growth Factor-1 (IGF-1), and Homocysteine (HCY)

Serum GH and IGF-1were assessed via commercially available enzyme-linked immunosorbent assay (ELISA) kits (DRG International, Inc., Springfield, NJ, USA), whereas HCY was examined using a fluorometric assay (Sigma-Aldrich, St. Louis, MO, USA) and both subsequently analyzed with a microplate reader and associated software (Infinite Pro 200 with i-control™, Tecan, Austria). Sample absorbance was read at a wavelength of 450 nm for GH and IGF-1, as well as an excitation/emission wavelength of 658 nm/708 nm for HCY. Moreover, unknown concentrations were determined by linear regression against known standard curves. The average intra-assay and inter-assay coefficients of variation (CV%) for GH were 0.35% and 1.36%, respectively. Likewise, the average intra-assay and inter-assay CV% for IGF-1 were 0.5% and 1.62%, respectively. Finally, the average intra-assay and inter-assay CV% for HCY were 5.38% and 7.01%, respectively. 

### 2.8. Statistical Analyses

Previous *a* priori power analysis determined that a total of 5 participants was necessary to achieve an anticipated Eta squared (η^2^) = 0.30 and power (1−ß) = 0.80 at α = 0.05. Additionally, all variables were tested for normality and homogeneity of variance using the Shapiro–Wilks test and Levene’s test of homogeneity of variance. The Greenhouse–Geisser correction was used in lieu of Mauchly’s test of sphericity due to the greater number of repeated measures relative to participant number, ultimately resulting in insufficient degrees of freedom. All anthropometric (body weight and BF%), dietary (relative protein, carbohydrate, fat, and fiber [g·kg^−1^]), hydration (ICW and ECW), as well as serum (PCV%, GH, IGF-1, HCY, and betaine) variables were assessed along the timeframe of the investigation (PRE, POST0, -4, -7, -10, -13, -16, -19, -22, -25, -28) via separate one-way analysis of variance (ANOVA) with repeated measures. Upon any significant ANOVA model main effect, pairwise comparison analyses were employed with a Bonferroni adjustment for alpha inflation. Eta squared (η^2^) was used to estimate the proportion of variance in the dependent variables explained by the independent variable. Eta squared effect sizes are determined to be: weak = 0.17, medium = 0.24, strong = 0.51, very strong = 0.70 [36]. Any dependent variables failing to meet normality and/or homogeneity assumptions were assessed using nonparametric Friedman’s ANOVA. Kendall’s W coefficient of concordance was used as an estimate of agreement for nonparametric data, whereby 0–0.19 = slight agreement, 0.20–0.39 = fair agreement, 0.40–0.59 = moderate agreement, 0.60–0.79 = substantial agreement, and >0.80 = almost perfect agreement [37]. Confidence intervals (CI) for significant comparisons are reported as 95% CI (lower bound, upper bound). Additionally, inter-rater and intra-rater reliability for DEXA body fat analysis was determined through intraclass correlation coefficient (ICC) analysis and their 95% CI, based on two-way mixed-effects models. Values <0.5, between 0.5 and 0.75, between 0.75 and 0.9, and >0.9 indicate poor, moderate, good, and excellent reliability, respectively [38]. All analyses were performed in SPSS V.27 (IBM Corporation; Armonk, NY, USA) at a significance level of *p* < 0.05 and values are reported as the means ± standard deviations (SD).

## 3. Results

### 3.1. Participant Descriptives and Body Composition Analyses

All participant descriptive data and anthropometric data are displayed in Table 1. Briefly, participant weight did not significantly change throughout the investigative timeframe. Analyses nonetheless discovered a significant time effect for BF% (*p* = 0.003; η^2^ = 0.816), whereby participants had a significantly lower BF% at POST28 relative to PRE (*p* = 0.006; CI [−3.241, −0.959]). Conversely, neither PRE (*p* = 0.691) nor POST28 (*p* = 0.069) were significantly different versus POST0. The intra- and-inter-class correlation coefficients for BF% were excellent (0.932; 95% CI [0.452, 0.933] and 0.976; 95% CI [0.713, 0.998]), respectively. All five participants additionally demonstrated 100% supplement compliance as per the turn of empty supplement containers and consistent verbal confirmation of protocol adherence. Notably, the participants consumed a range of 0.05–0.08 (0.07 ± 0.01) g·kg^−1^ betaine with respect to their PRE bodyweight.

### 3.2. Dietary Assessments

Participant relative macronutrient and fiber content are displayed in Table 2, whereby analyses failed to reveal any significant differences across baseline or any of the assessed post-supplementation time points. Dietary records and subsequent RD inspection revealed that two out of the five participants did not consume adequate folate/folic acid and thus were required to supplement with the B-vitamin complex which increased in folate/folic acid to normal levels. Collectively, participant dietary trends incidentally remained constant throughout the investigative timeline (see Table 2), ultimately dictating that those required to supplement with the B-vitamin complex did so until this study had concluded and vice versa for those with nutritionally adequate micronutrient intakes.

### 3.3. Hydration and Serum Analyses

All hydration and serum marker analyses are displayed in Table 3. Briefly, there were no significant differences across the investigative timeline for hematocrit (PCV%), nor in cellular compartment-specific (ICW and ECW) hydration. Furthermore, neither GH, IGF-1, nor HCY displayed any significant changes between baseline and any post-supplementation time point. 

Analyses revealed a significant “very strong” time effect for serum betaine concentrations (*p* = 0.010; η^2^ = 0.820). Nevertheless, data at the POST19 and POST25 time points violated normality assumptions, whereby nonparametric analyses confirmed the significant effect (*p* < 0.001; Kendall’s W = 0.653) with substantial agreement. Although no significant pairwise comparisons were observed due to Bonferroni α adjustments, a follow-up analysis was employed to further determine the post-supplementation time point at which serum betaine concentrations were no longer statistically significant from baseline values. Therein, a one-way ANOVA with repeated measures analyzing betaine concentrations across the PRE, POST0, POST4, and POST7 time points was selected upon visual data assessment and a Bonferroni correction once again employed (see Figure 2). A significant time effect was once again observed (*p* = 0.007; η^2^ = 0.842), demonstrating a significant elevation in serum betaine at POST0 relative to baseline (*p* = 0.047; CI [0.159, 17.505 µg·mL^−1^]). Additionally, serum betaine concentrations at POST4 and POST7 were not statistically different relative to baseline values.

## 4. Discussion

Several prior performance-oriented crossover investigations have previously implemented a range of betaine supplementation washout periods, utilizing limited pharmacokinetic data and without confirming full intervention reversibility [10,11,19,20,21,27]. The present pilot investigation sought to elucidate whether a 28 day period was sufficient to return serum betaine, total body water parameters, hematocrit, as well as serum GH, IGF-1, and HCY to baseline following 14 days of betaine supplementation. Aligned with our hypothesis, our data demonstrated an expected rise in serum betaine immediately following the supplementation period that declined to statistically nonsignificant levels at 4 days post-supplementation. The current findings also substantiate prior crossover designs that have cumulatively employed a washout timeframe range between 1 and 3 weeks [10,11,27]. Nevertheless, we did not anticipate such a precipitously reduced serum betaine given prior pharmacokinetic data on repeated dosing and its associated augmentations in half-life [20,21]. As Schwahn et al. [20] detailed an average elimination half-life increase from 14 to 41 h with five days of (50 mg·kg bodyweight^−1^) betaine ingestion, our group somewhat expected a longer reversal period to baseline serum betaine concentrations given our extended supplementation period and relatively higher generalized dosage. The current data otherwise demonstrate that our supplement protocol did not profoundly enhance serum betaine elimination half-life, evidenced by concentrations that were ~76% attenuated at POST4 relative to the initial average 8.79 µg·mL^−1^ increase from baseline (see Table 3). Furthermore, although the present findings would ultimately suggest that 4 days following betaine supplementation is statistically equivocal to pre-administration levels, we posit that the small participant pool employed in this pilot study significantly contributed to a disproportionate degree of variation and thus should be interpreted cautiously. The POST7 time point nevertheless is comparably nonsignificant and more visually indistinguishable (relative to mean ± SD) versus pre-supplementation values. Therefore, a more conservative approach would indicate that serum betaine concentrations both reliably and statistically return to baseline at 7 days post-supplementation.

Despite a relatively robust dosing regimen, several betaine-associated outcomes assessed in the present study were not significantly impacted. Specifically, HCY remained unchanged throughout the investigative timeframe despite substantial serum betaine augmentations. A previous meta-analysis by Deminice et al. [39] detailed that resistance—and not aerobic—exercise cumulatively results in attenuated HCY. Therefore, perhaps we failed to observe any significant HCY changes due to our training status criteria, synergistically combined with our participants’ relatively young age range (homocysteine is positively correlated with increasing age) and consistent individualized micronutrient standardization [39,40,41]. Furthermore, 14 days of betaine supplementation failed to statistically alter cellular water content or PCV%, regardless of its osmolytic—thus ostensibly hydrating—role [6,9]. The former equivocal finding is nonetheless corroborated by prior investigations in athletic populations, whereby we can similarly postulate that an inability to detect any ICW and/or ECW changes was potentially due to uncontrolled participant dietary habits [13,42]. Moreover, BIA total body water assessment may have lacked the measurement sensitivity necessary to analyze skeletal muscle hydration as the primary tissue of concern. Our laboratory has previously demonstrated that more precise methods such as tissue sample visualization via transmission electron microscopy likely provides a more precise inspection into intracellular fluid content [43]. While betaine supplementation has also ostensibly been credited to augment red blood cell and hemoglobin counts, our data failed to corroborate this notion via hematocrit (PCV%) analysis [6,7]. Soccer players supplemented 2 g/day betaine in an intervention by Nobari et al. [44] similarly did not display increased red blood cell or hemoglobin content, but retained red cell distribution width and mean corpuscular hemoglobin relative to placebo following a 14-week training timeframe. The authors therein posit that betaine may better function to protect against erythrocyte hemolysis rather than augmenting total associated cell counts [44]. 

Finally, it is prudent to note the statistically significant body composition improvements observed in the present study. Contrary to previous investigations that have demonstrated betaine supplementation-mediated body fat reductions and/or lean mass accretion, the current study did not detect any discernable reductions in the former until 28 days post-supplementation [12,13]. While the reason for this time point-specific improvement is largely unclear, it remains possible that the aforementioned small participant pool, as well as an inability to include a placebo arm and/or control individualized training prescriptions fundamentally exacerbated existing between-subject variation. Nonetheless, our methods attempted to account for discrepancies amongst nutrient status and week-to-week dietary intake and thus it is also possible that this finding holds speculative credibility. Prior authors have described that betaine supplementation may take up to four weeks before actualizing any associated benefits alongside concurrent resistance training [12]. Notwithstanding the extended timeframe whereby body fat percentage was reduced, perhaps the participants in the present study investigation gleaned performance-specific benefits during their supplementation period and during the onset of the washout period that were ultimately undetected until the final body composition assessment.

### Limitations

Several aspects of the current investigation are limited by the inherent characteristics of a pilot study design with an associated convenience sampling method. Most notably, the absence of a placebo condition alongside the aforementioned small participant pool largely inflates extraneous variation. Although we further attempted to maximize dietary control across the investigative timeframe, the presently employed 24 h recall methodology may not accurately represent long-term habitual patterns [45]. Nevertheless, these cumulative restrictions were otherwise warranted to construct a longitudinal supplementation protocol with extensive sampling time points and whilst curtailing attrition. We also affirm that the dietary recall frequency currently employed largely reassured that all participants had sufficient methyl metabolism-relevant micronutrient intakes; ergo, all nutrition-mediated detrimental impacts on betaine were minimized [7]. As previously mentioned, participant resistance training protocols were also unstandardized. Although betaine supplementation has previously been credited with augmenting resting serum GH and IGF-1 concentrations, the present study may have demonstrated equivocal results considering the former—and thus the latter by association—hormone is tightly correlated with exercise-mediated lactate levels [1,5,46]. Future research should consequently seek to substantiate our findings with a larger participant pool and randomized control arm, as well as implementing a standardized training prescription and periodic serum sampling to determine exercise-specific GH and IGF-1 alterations. 

Lastly, these data are wholly limited to illustrate the impact of 14 days betaine supplementation on serum concentrations. Notwithstanding its strong correlation with serum levels, the current investigation did not directly assess skeletal muscle betaine content [22]. Furthermore, this relationship has been demonstrated solely in rodents and is hitherto undescribed in a human model [22]. As our participants ostensibly displayed reduced overall HCY due to their resistance training history, age, and micronutrient standardization, it is also possible that a reduced requirement for enzymatic BHMT activity could result in a commensurately augmented muscle betaine saturation [9,39,40,41]. Consequently, this phenomenon may delay the true washout time as skeletal muscle concentrations continue to decline beyond serum values towards pre-supplementation levels. Provided the aforementioned strong correlation between skeletal muscle and serum betaine concentrations, the authors of the present study are nonetheless confident that a 7 day period is reasonably sufficient to return tissue levels to baseline provided the low concentrations observed of serum values at—and beyond—the specified POST7 time point [22]. Regardless, these postulations are otherwise tentative and thus warrant future investigations to comprehensively elucidate the impacts of a similarly constructed betaine supplementation protocol on skeletal muscle-specific washout timeframes.

## 5. Conclusions

The present study ultimately sought to determine an appropriate washout period following a 14 day betaine supplementation protocol. Our data therein demonstrate that a 7 day timeframe was sufficient to return serum betaine concentrations to pre-supplementation levels even during one of the highest reported loading doses in the sports nutrition literature [9]. Nevertheless, the current investigation employed a pilot study approach to substantiate prior and inform succeeding crossover designs [10,11,27]. Beyond betaine concentrations, we demonstrated several equivocal serum parameters and hydration indices that would also largely benefit from further investigation with an increased participant pool, as well as an exercise intervention. Regardless, these data thereby provide a precedent for future betaine-associated performance research, ostensibly facilitating future understanding of this promising supplement’s ergogenic potential.

## Figures and Tables

**Figure 1 nutrients-14-00498-f001:**
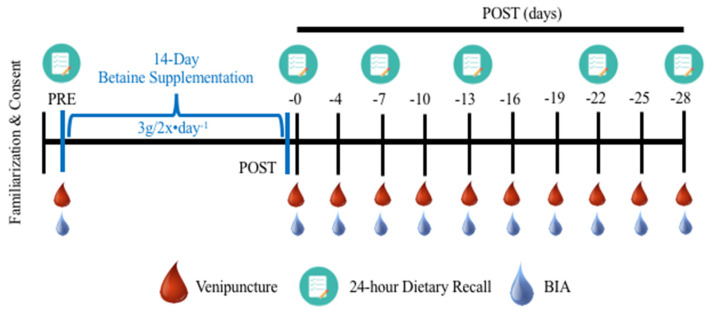
Visualization of investigation methodology and general timeline. BIA = bioelectrical impedance analysis.

**Figure 2 nutrients-14-00498-f002:**
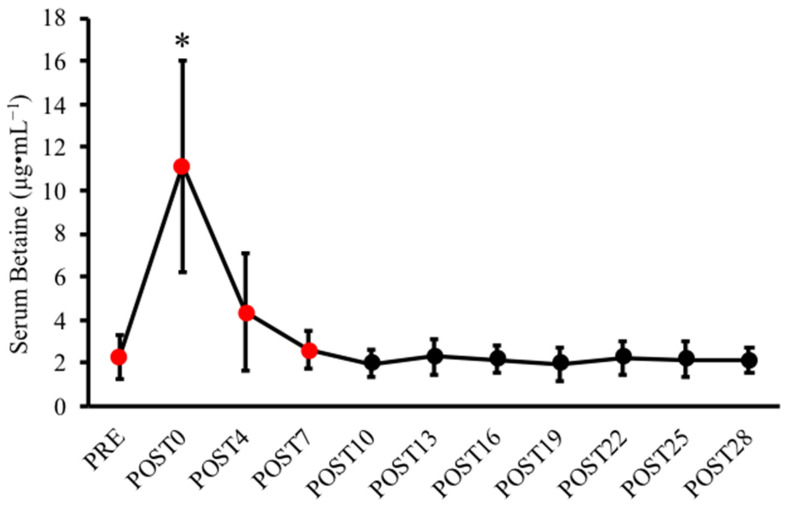
Serum betaine concentrations at baseline and during a 28 day washout period. All data are reported as the means ± SD and performed at a significance level of *p* < 0.05. * Significant time effect, whereby follow-up analyses (data points analyzed are depicted as red circles) determined a significantly elevated serum betaine at POST0 relative to PRE. Furthermore, neither POST4 nor POST7 concentrations were statistically different from baseline.

**Table 1 nutrients-14-00498-t001:** Participant demographics, anthropometrics, and body fat (BF%). All data are presented as the means ± SD and performed at a significance level of *p* < 0.05.

Mean ± SD	PRE	POST										*p*-Value; η^2^
*n* = 5		-0	-4	-7	-10	-13	-16	-19	-22	-25	-28	
Age (years)	26 ± 6											n/a
Training Age * (years)	7.8 ± 3.1											n/a
Height (cm)	171.7 ± 5.0											n/a
Weight (kg)	93.4 ± 15.7	93.2 ± 15.4	93.0 ± 15.6	93.2 ± 15.8	93.4 ± 15.7	93.1 ± 15.7	92.9 ± 15.8	93.2 ± 15.9	93.3 ± 15.8	93.1 ± 15.9	82.9 ± 16.0	0.655; 0.101
Body Fat (%)	18.6 ± 4.8	18.0 ± 4.6									16.0 ± 4.2 *	0.003; 0.816

Participant body weight was not significantly altered, but BF% was significantly decreased at POST28 versus PRE. * Training age is defined as the number of years each participant has historically met the resistance training inclusion criteria.

**Table 2 nutrients-14-00498-t002:** Participant dietary macronutrient and fiber, as well as methyl metabolism-associated micronutrient (vitamin B2, B6, folate, B12, and choline) intake for the 24 h preceding PRE, as well as POST0, -7, -13, -22, and -28 time points. Macronutrients and fiber are reported as average relative consumption (g·kg^−1^ bodyweight), whereas micronutrients are reported in absolute quantities. All data are presented as the means ± SD and performed at a significance level of *p* < 0.05.

Mean ± SD	PRE	POST					*p*-Value; η^2^
*n* = 5		-0	-7	-13	-22	-28	
PRO (g·kg^−1^)	1.81 ± 0.40	1.74 ± 0.53	2.21 ± 0.79	1.66 ± 0.63	1.86 ± 0.51	1.78 ± 0.53	0.356; 0.224
CHO (g·kg^−1^)	2.68 ± 0.82	3.26 ± 0.96	3.21 ± 1.00	3.08 ± 0.97	3.13 ± 1.12	3.46 ± 1.07	0.330; 0.243
FAT (g·kg^−1^)	0.92 ± 0.36	1.09 ± 0.29	1.19 ± 0.49	0.98 ± 0.29	0.93 ± 0.27	1.12 ± 0.28	0.498; 0.160
Fiber (g·kg^−1^)	0.28 ± 0.14	0.26 ± 0.11	0.34 ± 0.14	0.26 ± 0.12	0.33 ± 0.24	0.27 ± 0.10	0.624; 0.106
Micronutrient Intakes							RDA/AI
B2 (mg)	3.38 ± 3.18	3.72 ± 2.31	4.90 ± 2.46	4.51 ± 2.93	4.80 ± 3.73	2.68 ± 1.15	1.30
B6 (mg)	3.86 ± 2.55	5.16 ± 2.44	7.10 ± 3.49	4.67 ± 4.67	7.89 ± 5.78	4.15 ± 3.30	1.30
B12 (mcg)	12.34 ± 5.49	10.89 ± 5.49	12.68 ± 5.17	12.89 ± 4.15	14.06 ± 5.29	10.47 ± 4.08	2.40
Folate (mcg) *	435.31 ± 215.12	509.70 ± 279.47	509.93 ± 443.12	524.15 ± 146.79	501.93 ± 250.95	445.43 ± 205.02	400.00
Choline (mg)	435.31 ± 152.68	509.70 ± 444.30	509.93 ± 333.08	524.15 ± 303.70	501.93 ± 179.46	445.43 ± 239.74	550.00

CHO = carbohydrate; PRO = protein. * Folate listed is dietary folate equivalent. All micronutrient dietary intake standards are based on current (2020–2025) recommendations for males aged 19–30 years. In brief, no dietary changes were evident throughout the investigative timeline.

**Table 3 nutrients-14-00498-t003:** Raw hydration and serum analyte data across all baseline and post-supplementation time points. All data are presented as the means ± SD.

Mean ± SD	PRE	POST										*p*-Value;
*n* = 5		-0	-4	-7	-10	-13	-16	-19 *	-22	-25 *	-28	η^2^
PCV (%)	47.2 ± 1.8	46.8 ± 2.4	47.2 ± 1.8	47.8 ± 2.0	46.4 ± 2.3	46.6 ± 2.4	46.2 ± 2.5	47.0 ± 3.0	46.8 ± 2.3	47.4 ± 1.7	46.8 ± 2.3	0.540; 0.152
ICW (kg)	32.3 ± 2.7	32.8 ± 2.7	32.1 ± 3.2	32.8 ± 2.8	32.8 ± 2.9	32.7 ± 2.6	32.8 ± 2.9	32.7 ± 3.0	32.7 ± 3.1	33.0 ± 3.0	32.8 ± 2.9	0.521; 0.192
ECW (kg)	18.4 ± 1.5	18.7 ± 1.3	18.5 ± 1.5	18.8 ± 1.6	18.8 ± 1.7	18.7 ± 1.5	18.9 ± 1.7	18.8 ± 1.7	18.9 ± 1.7	18.8 ± 1.7	18.8 ± 1.5	0.328; 0.241
Serum Markers												
Betaine (µg·mL^−1^)	2.31 ± 1.05	11.1 ± 4.91	4.38 ± 2.71	2.61 ± 0.88	1.99 ± 0.66	2.29 ± 0.84	2.17 ± 0.60	1.98 ± 0.77	2.25 ± 0.80	2.17 ± 0.83	2.11 ± 0.60	0.010; 0.820
GH (ng·mL^−1^)	0.50 ± 0.77	0.09 ± 0.10	1.26 ± 1.81	1.51 ± 2.24	0.46 ± 0.76	0.26 ± 0.32	0.50 ± 0.61	0.27 ± 0.22	0.16 ± 0.28	0.15 ± 0.23	0.19 ± 0.29	0.279; 0.276
IGF-1 (ng·mL^−1^)	144. ± 65.4	136.2 ± 54.6	155.2 ± 66.5	173.8 ± 88.0	173.5 ± 81.0	179.5 ± 83.4	168.4 ± 74.0	171.9 ± 98.2	154.5 ± 62.2	147.0 ± 62.9	135.4 ± 53.2	0.226; 0.311
HCY (µmol·mL^−1^)	26.9 ± 7.4	22.9 ± 4.4	25.1 ± 5.7	26.4 ± 3.3	24.9 ± 3.4	25.7 ± 7.0	25.6 ± 6.3	28.2 ± 6.7	27.4 ± 12.2	28.5 ± 11.7	29.1 ± 7.5	0.597; 0.173

ECW = extracellular water; GH = growth hormone; ICW = intracellular water; HCY = homocysteine; IGF-1 = insulin-like growth factor-1; PCV = packed cell volume. * Notably, serum betaine at POST19 and POST25 violated normality assumptions and therefore the medians and interquartile ranges (Q3–Q1) were 1.64 (2.60–1.53) and 1.85 (2.88–1.62).

## Data Availability

The data presented in this study are available on request from the corresponding author.

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
