# Peer review of "Human Serum Betaine and Associated Biomarker Concentrations Following a 14 Day Supplemental Betaine Loading Protocol and during a 28 Day Washout Period: A Pilot Investigation"

_nutrients, 2022, doi:10.3390/nu14030498_

Round 1

Reviewer 1 Report

This is an interesting study but there are some areas that need improvement before I recommend the manuscript. I am unsure why you selected five participants for this study? Was it a convenience sample? There is also more information required about the participants such as training history because of the small sample size that will not allow any generalizing of results. Some more information regarding the rationale of the study design is needed. Why the 14-day loading, the dosage of betaine and following participants for 28 days? Please provide this information. Further comments are provided below.

Line 108: Why 28 days? Do you have any evidence that guided you to select this duration?

Lines 118-119: The symbols may be incorrect? Should it be equal to/greater than? Otherwise your participants could be relatively untrained. I would also suggest some more information about your participants in terms of resistance training and strength levels.

Lines 134-144: How do you know the participants had taken the supplements? What was the adherence rate?

Lines 161-163: For the DEXA scan were participants scanned with minimal clothing, after an overnight fast and hydration status assessed (e.g. urine specific gravity)? Who scanned the participants? What was the intra/inter-tester reliability in your laboratory?

Lines 265-266: The threshold need slight adjustments because is 0.40 slight or fair agreement? 0.60 fair or substantial agreement?

Line 269: Data that is not normally distributed should be reported as median and interquartile range (IQR).

Line 279: How was compliance assessed?

Line 281-282: The results should not be reported in the title for Table 1. The symbols plus footnotes should be sued to explain significant findings.

Lines 297-298: These lines should be in the footnotes of Table 2.

Lines 324: ECW and ICW.

Line 348: What was the relative dosage of betaine for the participants in the study?

Reviewer 2 Report

Machek et al. provides an intriguing pilot study about 14-day supplementation of betaine to healthy resistance trained men, taking it anhydrous at 3 g / 2 times per day, and the subsequent timeframe for washout by collecting 10 samples by venipuncture until 28 days after the last supplement dose. This study answers currently unknown estimations about when betaine is cleared after the dose has been stopped. The authors found that 4 days after supplementation had ceased, then the serum betaine values reached baseline values.

Comments:

  • Please explain why you chose only 5 subjects to do this analysis? Why did you not choose 10 or 15? Your standard deviations are quite high at Post0 in Figure 2 and also in Table 3. For a study to be published in a reputable journal, I would suggest that you perform a power calculation such that you can demonstrate possibly meaningful results. Your study methods were proper, but I would do all I could to raise the n here. Or, I would elaborate more in the Discussion section to answer any questions a reader may have.
  • Can you elaborate more on the type of resistance training that the subjects did and why you chose this parameter for your study question? According to the ACSM recommendations you stated, it seems that these subjects may not be the best choice for an interest in ergogenic potential.
  • Spelling mistake in Table 3 “Makers” to “Markers.”  

Round 2

Reviewer 1 Report

Well done on addressing my concerns and improving the quality of your paper.

Reviewer 2 Report

Thank you for your revisions and explanations.